# Identification of cholera hotspots in Zambia: A spatiotemporal analysis of cholera data from 2008 to 2017

John Mwaba[1], Amanda K. Debes[2], Patrick Shea[2], Victor Mukonka[3], Orbrie Chewe[3], Caroline Chisenga[1], Michelo Simuyandi[1], Geoffrey Kwenda[4], David Sack[2], Roma Chilengi[1], Mohammad Ali[2]*

**1** Centre for Infectious Disease Research in Zambia, Lusaka, Zambia, **2** Johns Hopkins Bloomberg School of Public Health, Baltimore, Maryland, United States, **3** Zambia National Public Health Institute, Lusaka, Zambia, **4** University of Zambia, School of Health Sciences, Lusaka, Zambia

* mail25@jhu.edu

## Abstract

The global burden of cholera is increasing, with the majority (60%) of the cases occurring in sub-Saharan Africa. In Zambia, widespread cholera outbreaks have occurred since 1977, predominantly in the capital city of Lusaka. During both the 2016 and 2018 outbreaks, the Ministry of Health implemented cholera vaccination in addition to other preventative and control measures, to stop the spread and control the outbreak. Given the limitations in vaccine availability and the logistical support required for vaccination, oral cholera vaccine (OCV) is now recommended for use in the high risk areas ("hotspots") for cholera. Hence, the aim of this study was to identify areas with an increased risk of cholera in Zambia. Retrospective cholera case data from 2008 to 2017 was obtained from the Ministry of Health, Department of Public Health and Disease Surveillance. The Zambian Central Statistical Office provided district-level population data, socioeconomic and water, sanitation and hygiene (WaSH) indicators. To identify districts at high risk, we performed a discrete Poisson-based space-time scan statistic to account for variations in cholera risk across both space and time over a 10-year study period. A zero-inflated negative binomial regression model was employed to identify the district level risk factors for cholera. The risk map was generated by classifying the relative risk of cholera in each district, as obtained from the space-scan test statistic. In total, 34,950 cases of cholera were reported in Zambia between 2008 and 2017. Cholera cases varied spatially by year. During the study period, Lusaka District had the highest burden of cholera, with 29,080 reported cases. The space-time scan statistic identified 16 districts to be at a significantly higher risk of having cholera. The relative risk of having cholera in these districts was significantly higher and ranged from 1.25 to 78.87 times higher when compared to elsewhere in the country. Proximity to waterbodies was the only factor associated with the increased risk for cholera (P<0.05). This study provides a basis for the cholera elimination program in Zambia. Outside Lusaka, the majority of high risk districts identified were near the border with the DRC, Tanzania, Mozambique, and Zimbabwe. This suggests that cholera in Zambia may be linked to movement of people from neighboring areas of cholera endemicity. A collaborative intervention program implemented

**Data Availability Statement:** All relevant data are within the manuscript and its Supporting Information files.

**Funding:** Funding support for data analysis and report preparation was provided by Bill and Melinda Gates Foundation through the Delivering Oral Vaccine Effectively (DOVE) project, administered by the Johns Hopkins Bloomberg School of Public Health (OPP1148763). The funder of the study had no role in study design, data collection, analysis, or interpretation, or writing of the report.

**Competing interests:** The authors have declared that no competing interests exist.

in concert with neighboring countries could be an effective strategy for elimination of cholera in Zambia, while also reducing rates at a regional level.

## Author summary

Zambia has experienced cholera outbreaks since 1977. It is a landlocked country bordered by the DRC and Tanzania to the north, Malawi and Mozambique to the east and Zimbabwe to the south; all of which experience regular cholera outbreaks. The Zambian Ministry of Health included cholera vaccination, in addition to standard cholera control measures, e.g., clean water, improving sanitation and promoting hygiene to counter a cholera outbreak in 2016. The implementation of these control measures is in line with Zambia's National Cholera Eliminating Plan (NCEP) by 2025 and is also consistent with guidance by the Global Task Force on Cholera Control's (GTFCC) global roadmap to end cholera by 2030. In both plans, the identification of high risk areas known as cholera "hotspots" is necessary to prioritize OCV deployment while also key in identifying areas where improvements are needed including surveillance systems and effective WASH improvements. In this study, we retrospectively analyzed district-level cholera data from 2008 to 2017. Sixteen of 72 districts were identified to have an increased risk of cholera using a geostatistical model. Outside of Lusaka district, which is a primary hotspot, the additional hotspot districts share borders with Zambia's neighboring countries. To achieve cholera elimination in Zambia by 2025, a regional strategy involving each of the countries bordering will be needed.

## Introduction

The global burden of cholera is increasing, with current estimates indicating that 1.3 billion people are at risk in endemic countries, resulting in 2.8 million cases and 91,000 deaths annually [1]. Of these cases, the majority (60%) occur in sub-Saharan. In Zambia, widespread cholera outbreaks have occurred since 1977 [2], predominantly in the capital city of Lusaka [3]. The causes have been attributed to poor access to safe water and sanitation facilities in peri-urban areas of the city [2]. To prevent and control cholera outbreaks, the Zambian government has adopted a multi-sectorial approach that engages relevant ministries and cooperating partners, as was the case for the 2017–18 outbreaks [4]. The intervention program includes provision of adequate safe water, improving sanitation facilities, and vaccinating individuals in Lusaka. Provision of adequate safe water and improving sanitation facilities are long-term measures which tend to be extremely expensive, require infrastructural change, skilled personnel for implementation, and management of the infrastructure [5]. As a short-term measure, vaccination programs will be implemented to control the disease [6,7].

A successful vaccination program requires a well-designed implementation plan. The World Health Organization (WHO) has advised the use of the oral cholera vaccine (OCV) in areas that are deemed at high risk or "hotspots" [8,9]. Since cholera has a spatial expression, understanding the geographical distribution of the disease is important for implementation of an effective intervention strategy [10]. Importantly, spatiotemporal clusters of cholera should be identified, i.e., areas where cholera incidence is significantly higher and occurs more frequently than elsewhere in the country. Identification of these areas provides the needed information to allocate resources to intervention programs targeted to these sites [11].

In the past, the OCV programs in Zambia lacked in-depth understanding of areas of high transmission. The aims of this study were to identify areas with an increased risk of cholera in space and time, and to perform an area-based analysis to understand the driving factors for the risk of cholera in these areas. This knowledge will help in developing an effective intervention strategy for controlling cholera in Zambia.

## Materials and methods

### The study area

Zambia is a landlocked country in Southern Africa and is located between latitudes 8˚ and 18˚ south and longitudes 22˚ and 34˚ east of the equator covering a total area of 752,612 km$^2$ [12]. The country is surrounded by, Malawi to the east, Mozambique, Zimbabwe, Botswana and Namibia to the South, Angola to the west, The Democratic Republic of Congo (DRC) to the north, and Tanzania to the north-east. The country is divided into 10 administrative provinces encompassing a population of approximately 16.6 million people as of 2016, and an estimated annual growth rate of 3.0 percent. The country further divided administratively into 114 districts as of August 2018. For the purposes of this analysis, we restricted the analysis to the 2010 census with 72 districts.

### Cholera data

This analysis included cases per WHO criteria: any patient aged 5 years or more presenting with acute watery diarrhea and severe dehydration where cholera is not known to be occurring, or any patient 2 years or older presenting with acute watery diarrhea where cholera is known to be occurring. A suspected case in which *Vibrio cholerae* 01 or 0139 was isolated from stool is considered a confirmed case [13]. This analysis includes cases reported by year and by district from 2008 through 2017; the data were obtained from the Zambian Ministry of Health Department of Public Health and disease surveillance database [14].

### Population and socioeconomic data

District level population and urban/rural proportion of population by district were obtained from the 2010 Census of Population and Housing report compiled by the Zambian Central Statistics Office (CSO) (http://www.mcaz.gov.zm/wp-content/uploads/2014/10/2010-Census-of-Population-Summary-Report.pdf). Additional socioeconomic data such as the percentage of the population living below the poverty index were obtained from the CSO Living Conditions Monitoring Survey 2015 [15].

### Water, Sanitation, and Hygiene (WASH) data

Data focused on access to improved sanitation and improved water sources was obtained from the Zambia Demographic and Health Survey 2014 (https://dhsprogram.com/what-we-do/survey/survey-display-406.cfm). The percentage of population using improved water source was defined as the population whose main source of drinking water was piped household water, a public tap or standpipe, tube-well or borehole, protected dug well, protected spring, collected rainwater, or bottled water. The percentage of the population with access to improved sanitation was defined as households with flush toilets, ventilated improved pit latrines, pit latrines with slabs, or composting toilets not shared with other households. The data were presented based on percentages for urban and rural population in the report. For this analysis, the data by district were calculated using district level urban/rural population percentages.

## GIS data

The digital maps of Zambia were obtained from The Humanitarian Data Exchange (https:// data.humdata.org/dataset/zambia-administrative-boundaries-level-1-provinces-and-level-2-districts-with-census-2010-population), which is shared under CC-by license (https://data.humdata.org/about/license).Until 2013, Zambia was subdivided into 72 districts. Since we have the data based on those 72 districts, we collapsed the 115 districts into those 72 districts for analysis. We compiled the cholera data and the other data sets in this study per district in the GIS database.

## Hotspots identification

We used a spatial scan test [16] to identify spatiotemporal hotspots of cholera from 2008 to 2017 in Zambia. A discrete Poisson-based space-time scan statistic was utilized to account for variations in cholera risk across both space (districts) and time (year) during the 10-year study period. Under the Poisson model, it was assumed that the number of cases for each segment of the study area would be proportional to the population, thus the model compared cases against the underlying population at risk. Since the location and size of the window changed in this process, the model created several distinct windows, therefore, a likelihood ratio was calculated. Under the Poisson model, the likelihood function for a specific window is:

$$\lambda = \left(\frac{n}{\mu}\right)^n \left(\frac{N-n}{N-\mu}\right)^{N-n} I\left(n > \mu\right)$$

where, $N$ is the number of cases in the study area, $n$ is the number of cases within the window, $\mu$ is the expected number of cases within the window under the null hypothesis, and $I()$ is an indicator function. The likelihood function was maximized over all windows, identifying the window that constituted the most likely cluster. The most likely cluster (hotspots) is the area that is least likely to have occurred by chance. The likelihood ratio for the window was noted and constituted the maximum likelihood ratio test statistic. Its distribution under the null hypothesis and its corresponding p-value was determined by repeating the same procedure on a large number of random replications of the data set generated under the null hypothesis using a Monte Carlo simulation approach.

In this study, since we were interested in the space-time scan statistic, the approach uses a cylindrical scanning window with a circular spatial base and height corresponding to time [16]. We set the spatial window to 20% of the population at risk assuming that a larger spatial window would obscure local details. In contrast, a smaller window would make the cluster individualistic in nature. We set the temporal window to 50% of the study period. We sought to identify the high-risk clusters, i.e. the areas where the interior of the scanning window are at a higher risk than the areas surrounding the window. The completion of the scan results in the identification of districts in which the risk of cholera was higher than the rest of the country during the study period. These high-risk districts represent cholera hotspots.

## Statistical analysis of the potential risk factors

**Zero-inflated negative binomial (ZINB) model.** To examine the potential drivers of cholera in Zambia, we first employed a zero-inflated negative binomial (ZINB) model [17] considering that the model would account for over dispersion and zero-inflation in the data set. The model assumes that our dataset contains two groups: a count regression group and an excess zero group. The count regression model fits the count data and the binary regression model fits the excess zero data. For each observation with probability p, the possible response

of the "excess zero group" is 0 count, and with probability of 1-p, the response of the count regression group is governed by a negative binomial with mean count of cases λ. If the response Y (cumulative number of cases over the study period) follows a ZINB distribution, then

$$
P(Y = y) = \begin{cases} p + (1-p)\left(\dfrac{k}{\lambda + k}\right)^{k}, & \text{if } y = 0 \\[2ex] (1-p)\dfrac{\Gamma(Y+k)}{\Gamma(k)\Gamma(Y+1)}\left(\dfrac{k}{\lambda + k}\right)^{k}\left(1 - \dfrac{k}{\lambda + k}\right)^{Y}, & \text{if } y > 0 \end{cases}
$$

where $0 \leq p \leq 1$, k is the overdisperson parameter and $\Gamma$ is the gamma function. We therefore modelled the ZINB regression as

- for count model: Log $\lambda = \beta_0 + \beta_1 x_1 + \beta_2 x_2$

- for excess zero model: Logit (p) = $\gamma_0 + \gamma_1 z_1 + \gamma_2 z_2$

where $x_i$ and $z_i$ are the variable of interest, and $\beta_i$ and $\gamma_i$ are the corresponding regression and zero-inflated coefficients, respectively. $\beta_0$ and $\gamma_0$ are the intercepts and logit (p) = log (p/1-p).

**Spatial dependency test.** We employed global Moran's *I* to test for spatial dependency of cholera in Zambia. The Moran's *I* was calculated as

$$
I = \frac{\sum_{i=1}^{m} \sum_{j=1}^{m} w_{ij}(r_i - \bar{r})(r_j - \bar{r})}{w_{ij} \sum_{i=1}^{m}(r_i - \bar{r})/m}
$$

where $r_i$ is the rate in region *i*, $r_j$ is the rate in region *j*, $w_{ij}$ is a measure of adjacency between region *i* and *j*, and is defined as (1 if *i* and *j* are adjacent; 0 otherwise). When rates in nearby areas are similar, the Moran's *I* will be large and positive, and when rates in nearby areas are dissimilar the Moran's *I* will be negative.

**Spatial regression.** It is important to note that spatial data may show spatial dependence in the variables and error terms, as the data collection using spatial units may reflect measurement error. This is because the administrative boundaries do not necessarily reflect the underlying process of disease transmission and the spatial dimension of the socioeconomic characteristics is an important aspect of the phenomenon. Therefore, based on the diagnostic test of the OLS, we further created spatial lag model (SLM) and spatial error model (SEM) to get the unbiased estimates of the factors for higher risk of cholera after adjusting for spatial heterogeneity of the outcomes and/or residuals. The SLM is defined as

$$
y = \rho w y + \beta x + \varepsilon
$$

where $\rho$ is the spatial lag parameter, and *wy* is the weighted average of its value in its neighborhood:
And, the (SEM) is defined as

$$
y = \beta x + \varepsilon, \text{ with } \varepsilon = \lambda w \varepsilon + \zeta
$$

Here, $\lambda$ is the spatial autoregressive parameter and the error $\zeta$ is independently and identically distributed. In case SLM, it is assumed that the observations are spatially dependent, whereas SEM assumes that the residuals are correlated with the neighborhood. Both SLM and SEM are estimated by maximizing the corresponding likelihood functions.

**Software applications.** We used SatSCan (https://www.satscan.org/) for identifying hotspots, Geoda (https://geodacenter.github.io/) for spatial analysis, SAS 9.4 for analyzing the data using ZINB model, and ArcMap Desktop 10.6 (Esri Inc.) for mapping of the hotspots.

**Ethics.** The study used secondary data aggregated at the district level, and the data analyzed were anonymized. The Ministry of Health, Zambia gave permission to access the data from the Department of Public Health and disease surveillance database. Therefore, no ethical approval was required for conducting this study.

## Results

In total, 34,950 cases of cholera were reported in Zambia between 2008 and 2017. The highest number of cases, 17,348, spanning 33 districts (almost half of the country) were reported in 2010. However, 89% of the cholera cases were reported from the Lusaka district in both 2009 and 2010. The lowest number of cholera cases (31 cases) were in 9 districts in 2015 (Fig 1).

Cholera cases also varied spatially by year (Fig 2). Starting with only a few districts affected near the DRC in 2008, cholera spread to a larger area from 2009–2012. Subsequently, the number of cases decreased from 2013–2017. Throughout the study period, Lusaka District had the highest burden of cholera, with 29,080 total reported cases.

The spatiotemporal analysis based on the district centers as the geographic coordinates yielded 16 high-risk clusters. Cholera hotspots were defined based on the location ID provided by the SatSCan for the identified clusters where 16 districts were found to be at a significantly higher risk of having cholera. The risk of having cholera in these districts ranged from 1.25 to 78.87 times compared to that elsewhere in the country (Fig 3).

About 4.7 million people (36% of the total population) live in these districts (Table 1).

We noted that although cholera occurred during several years in some districts, most were not identified as being significantly high-risk areas of cholera in the spatiotemporal analysis.

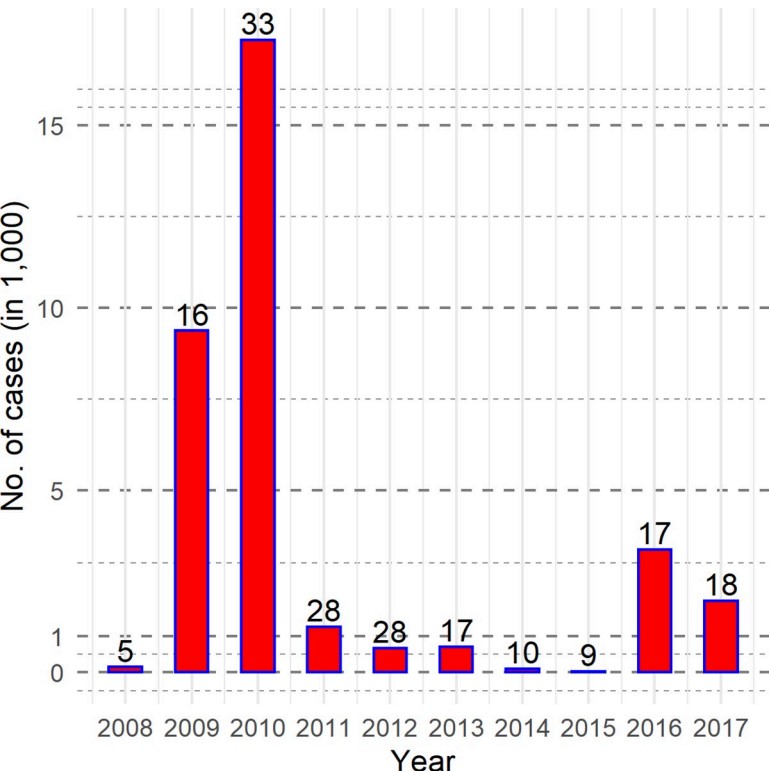

**Fig 1. Distribution of cholera cases, by WHO case definition, by year, 2008–2017.** Note: No. of cholera affected districts are recorded on the top of bars.

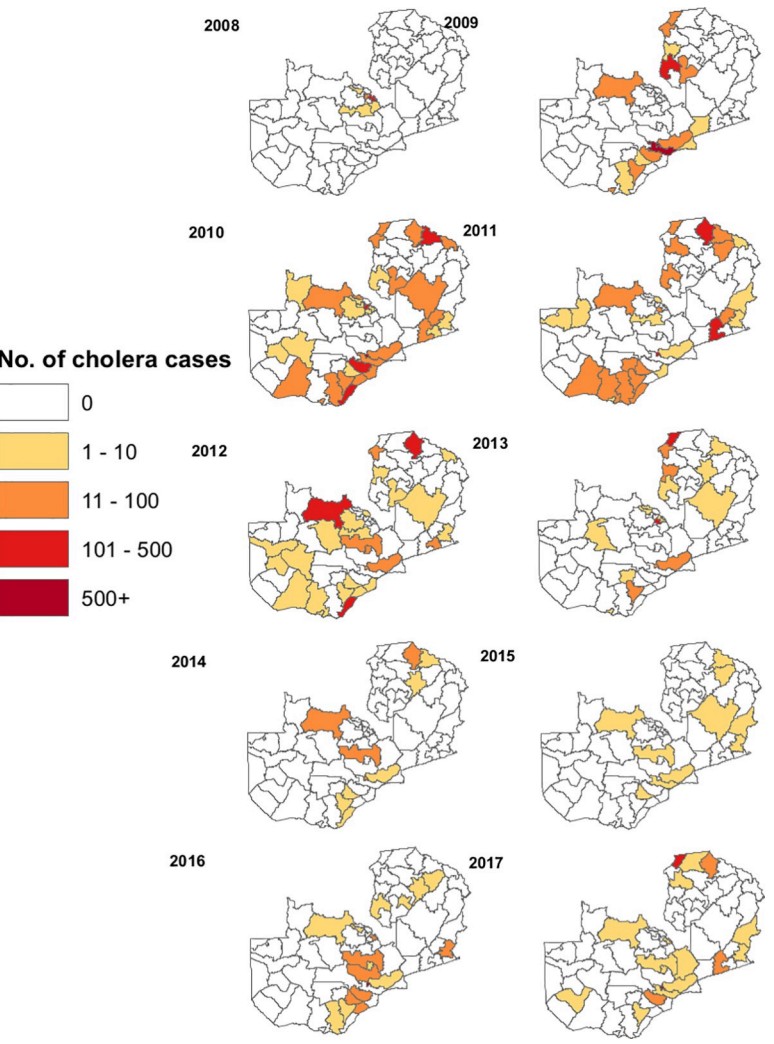

**Fig 2. Cases of cholera by district and by year, 2008–2017.**

For instance, Chongwe district experienced cholera 9 out of 10 years, but was not a hotspot in this model. Similarly, Choma district was affected 8 times, but not determined to be a hotspot. This is because there were fewer than 100 cases in these two districts during the study period, thus the relative risk is too low to be defined as a hotspot.

It is also notable that cholera did not affect entire districts but rather, only affected some parts. For instance, 3 of the 33 wards in Lusaka district reported over 50 cases of cholera per 100,000 population in three years (2016–2018) (Fig 4). 322,198 people live in these three wards compared to 1,747,152 people in all of Lusaka district. This indicates that only 18% of the population in the district were at a higher risk for cholera. It is important to note that despite Kabulonga ward being a low density area, this ward includes Bauleni compound which is highly populated area that experienced recurrent cholera outbreaks; hence the ward was determined to be a hotspot.

The descriptive statistics of the variables included in the risk factor analysis are presented in the Table 2.

After being adjusted for the incidence rate of first order neighborhoods, the ZIP model determined that "Distance from household to the nearest waterbody" was associated with

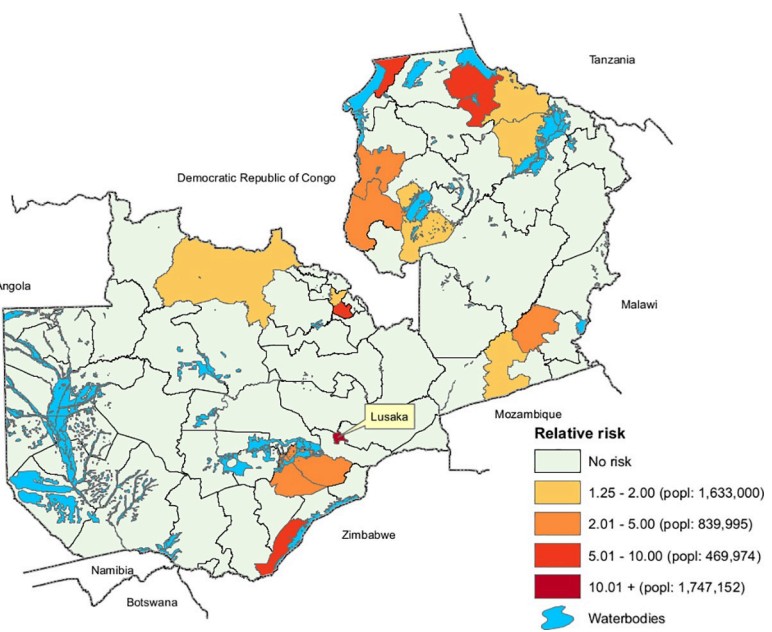

**Fig 3. Spatiotemporal hotspots of cholera in Zambia, 2008–2017.**

increased risk of cholera in Zambia (P < .05) (Table 3). Since, no other variables showed an association with risk for cholera we did not conduct a multivariable analysis.

The estimated Moran's I statistic showed a moderately significant spatial dependence (p = 0.09) in cholera incidence (Fig 5), suggesting exploration of spatial regression.

Diagnostic tests for the OLS models were conducted and the value of the Lagrange Multiplier was not statistically significant for either lag model (value: 0.52, p = 0.47) or error model (value: 1.86, p = 0.17). However, the Robust Lagrange Multiplier was found to be statistically significant for both the lag (value: 7.04, p = 0.01) and error model (value: 8.38, p = 0.003), indicative of conducting both SLM and SEM models. The results from OLS, SLM and SEM describing the effect of the factors on the outcome are presented in Table 4. Based on the model diagnostics and comparing the Akaike Information criterion (AIC) (the lower the better) and R-square (the higher the better), the SEM was found to be the best-fit model for the data. This was also supported by the lag coefficient (lamda = 0.40) and its associated p-value (0.0003) of the SEM model. However, none of the variables included in the spatial regression models were found to be associated with cholera incidence in the country. Importantly, we had a very high multicollinearity number (61136) in this analysis, indicative of highly correlated data for the independent variables included in the analysis.

**Table 1. Number of districts and population by risk group in Zambia.**

| Risk group | Relative Risk | Number of Districts | No. of Population | Percent of Total Population |
|---|---|---|---|---|
| Extremely high | 10.01+ | 1 | 1,747,152 | 13.34 |
| High | 5.01–10.00 | 4 | 469,974 | 3.59 |
| Medium | 2.01–5.00 | 5 | 839,995 | 6.42 |
| Low | 1.25–2.00 | 6 | 1,633,000 | 12.47 |
| **Total** | | **16** | **4,690,121** | **35.82** |

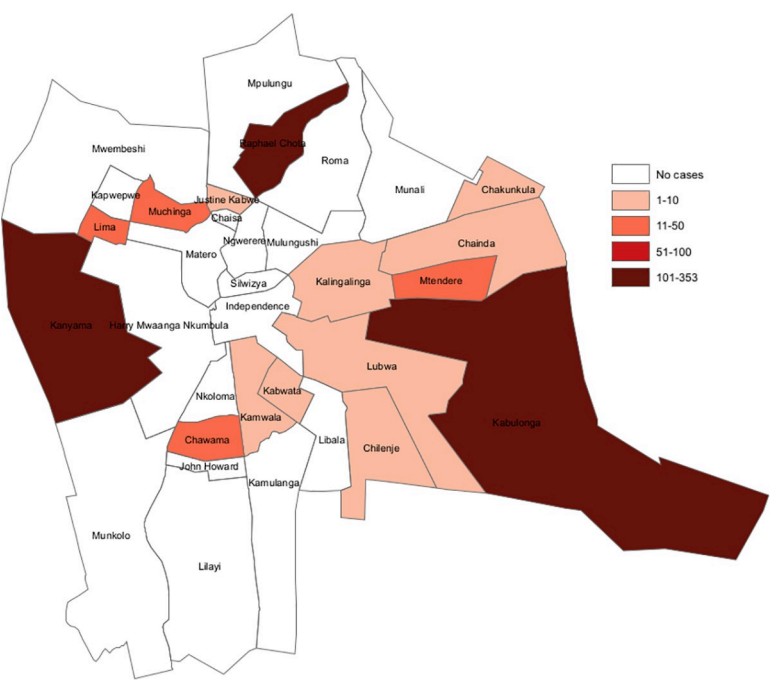

**Fig 4. Cholera cases in Lusaka by ward, 2016–2018.**

## Discussion

The results of our study identified 16 districts of Zambia as hotspots, but at varying levels of risk. Five of these districts had a relative risk >5. A major hotspot identified was the city of Lusaka; 89% of the cases in this analysis were reported from Lusaka. Interestingly, only 3 of 33 wards in Lusaka district were identified as high risk areas. The sub-district (constituency) level data analysis of Lusaka found that only three of seven constituencies, with about 20% of the population (600,000 people), experienced high rates of cholera. This suggests that control efforts should focus in these constituencies. Lusaka has several densely populated peri-urban settlement areas with inadequate water and sanitation infrastructure which compromises sanitation and hygiene, facilitating cholera transmission [3]. Lusaka has experienced prolonged rainfall that results in flooding, and this likely further increases cholera risk [2]. The city is also a center for an international cross-borders trading, with many people traveling between countries which may also increase risk of transmission to the city.

The hotspot areas outside Lusaka were near the borders with DRC, Tanzania, Mozambique, and Zimbabwe. Given the geographical distance between these border area hotspots, control

**Table 2. Descriptive statistics of the study variables (n = 72 districts).**

| Variable: | Mean | Median | Standard Deviation | Minimum | Maximum |
|---|---|---|---|---|---|
| Total population | 181,843 | 135,825 | 208,429 | 24,304 | 1,747,152 |
| Total number of cholera cases 2008–2017 | 486 | 20 | 3,397 | 0 | 29,080 |
| Population living in the urban area (%) | 25.47 | 13.50 | 28.45 | 2.02 | 100.00 |
| Households having access to improved sanitation (%) | 37.73 | 31.75 | 14.22 | 26.01 | 75.00 |
| Household having access to improved water source (%) | 57.52 | 52.81 | 12.94 | 21.51 | 90.00 |
| Households living under poverty (%) | 46.32 | 51.67 | 12.72 | 13.00 | 56.80 |
| Distance from the center of district to the nearest waterbody (km) | 26.33 | 21.41 | 12.81 | 13.00 | 56.80 |

**Table 3. Results of the analysis using zero inflated negative binomial model (ZINB) model.**

| Variables | Estimate | Wald 95% CI | P-value |
|---|---|---|---|
| Percent of population living in the urban area | 0.0028 | -0.0145 to 0.0201 | 0.75 |
| Percent of households having access to improved sanitation | 0.0056 | -0.0289 to 0.0401 | 0.75 |
| Percent of household having access to improved water source | 0.0213 | -0.0310to 0.0525 | 0.26 |
| Percent of households living under poverty | -0.0062 | -0.0449 to 0.0324 | 0.75 |
| Distance from center of the district to the nearest waterbodies | -0.0170 | -0.0338 to -0.0003 | 0.045 |

Note: Each variable was entered in the model in combination with the neighborhood incidence rate to adjust for the spatial structure of the disease. Only the negative binomial component of the model is provided, since the zero inflated component of the model did not converge.

efforts will be challenging. More localized analysis may likely reveal that the hotspot areas in these districts only encompass a select few wards, as was the case in Lusaka. The hotspots near the borders suggest that cholera in Zambia is linked to cross-border movement between countries where cholera is also endemic. This was observed in Uganda [18]. Further, this suggests that a collaborative intervention program with the neighboring countries could be an effective strategy to eliminate cholera in Zambia and a step toward reduction and elimination in the region.

The peak of cholera in Zambia was observed between 2009 and 2010, at beginning of the study period. Subsequently, the number of cases declined. Some have hypothesized that the large number of cases in Lusaka in 2009 and 2010 might be due to prolonged rainfall and flooding (https://www.who.int/cholera/countries/ZambiaCountryProfile2011.pdf), however, this time period was also a peak time for cholera in other African countries suggesting that the factors responsible for the high numbers may have occurred more generally in Africa. As depicted in Fig 2, cholera occurred sporadically in Zambia; thus, it was difficult to ascertain which risk factors would be the best predictors for its occurrence, rendering it difficult to identify any district level predictors for the increased risk of cholera in Zambia.

Interestingly, Chiengi and Mpulungu districts were identified as the areas of highest risk after Lusaka; however, neighboring Kaputa reported only 2 cases over the 10-year period. Of the highest risk districts after Lusaka (Chiengi, Mpulungu, and Sinazongwe), none reported cases in more than 5 out of the 10 years studied. In contrast, Solwezi district, a hotspot of lesser risk, reported cases in 8 of the 10 years. Since we did not identify any risk factors for cholera from our district level data except distance to the waterbodies, it is difficult to determine the

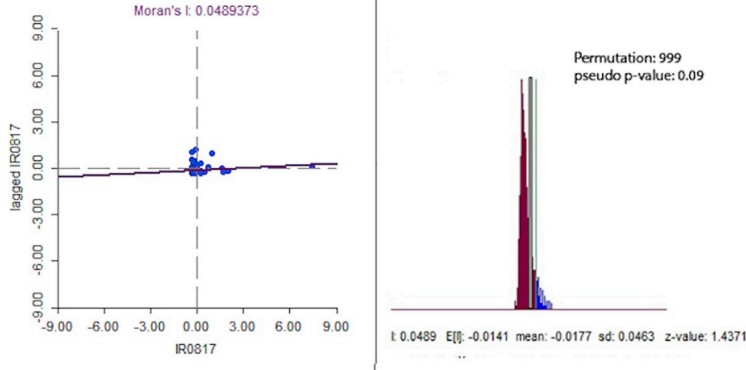

**Fig 5. Moran's I statistic and associated p-values based on 999 permutations.**

**Table 4. Results of the different regression analysis.**

| Variables | OLS | SLM | SEM |
|---|---|---|---|
| Population living in the urban area | -20.51 (0.63) | -21.84(0.60) | -35.49 (0.33) |
| Households having access to improved sanitation | 32.41 (0.67) | 28.72(0.69) | 38.74 (0.55) |
| Household having access to improved water source | 0.016 (0.82) | 0.0136(0.84) | 0.0084 (0.89) |
| Households living under poverty | -9.66(0.90) | -16.77 (0.82) | -36.12 (0.59) |
| Distance from waterbodies | -0.00057(0.96) | -0.0014 (0.90) | -0.0067 (0.53) |
| Multicollinearity condition number | 61136 | - | - |
| Lag coefficient | - | 0.18 (0.23) | 0.40 (.0003) |
| Akaike Information Criteria (AIC) | 318.371 | 319.562 | 314.778 |
| R-square | 0.0906 | 0.1074 | 0.1691 |

OLS = Ordinary Least Square regression, SLM = Spatial Lag regression Model, SEM: Spatial Error regression Model

underlying cause(s) of cholera. Note that increasing risk among people living proximate to water bodies has already been documented in a number of studies [18–20]. Since the best fit model in our study, i.e. SEM, explained only 16% of the total variations in the outcome, it is reasonable to believe that other risk factor(s) play a role at a spatial scale in Zambia.

Since cholera is transmitted by fecal oral route through water, one expects a relationship between cholera and WaSH conditions [21–24]. It is assumed that a well-managed, improved WaSH infrastructure, as has proven effective in industrialized countries, would be an optimal strategy for the cholera elimination program in Zambia. This study did not find an association between cholera and WaSH at the district level, the data available may not have been enough discriminating to find such an association. With the limitation of resources for major improvements in infrastructure, household level WaSH programs have been conducted in the past and might be effective [25–26]. While large-scale WASH interventions may ultimately eliminate cholera, cholera vaccination can be used in the interim as an effective control measure targeting the identified hotspots.

This study has limitations. By conducting the analysis based on the 72 districts with the available census population data, rather than the current 115 districts, there was a loss of spatial resolution in the identification of hotspots. Secondly, we did not include acute watery diarrhea cases less than 5 years old from areas where cholera was not known to have occurred or acute watery diarrhea cases less than 2 years old from areas where cholera was known to have occurred. Therefore, we could have missed some cholera cases in our study. Cholera can occur in these age groups and the decision to not include these may have led to an underestimation of cholera cases. Thirdly, the data was obtained from routine surveillance system and there could be differences in the reporting of cases from different parts of the county leading to reporting bias. The data on water and sanitation (WASH) were available only by urban and rural areas; thus there was limitation in the ability to calculate association of cholera with WASH. Also, the WASH data was only for a single time point, precluding an ability to perform time-series analysis with the data. Further, there was limited risk factor data available at the district level that could be used in our analysis, thus, we were unable to identify key risk factors for cholera as well as in predicting future outbreaks.

While accepting these limitations, this analysis has identified the districts with elevated risk of cholera which will facilitate the selection of sites for more intensive control strategies. By targeting the highest risk districts, as identified in our analysis, further investigation and data collection at the sub-district level are needed to identify the specific areas to be targeted for the interventions within each district. This would facilitate the planning of interventions in the

highest risk wards in a more cost-effective manner. For this, local participation and knowledge are needed to identify data and refine the analyses to highlight ward level high-risk areas within the districts. In the future, if very sensitive and specific surveillance methods allow for real-time case detection with GIS coordinates of cases, improved maps can be created, allowing for even better targeting of interventions.

The WHO announced the cholera elimination by 2030 program by partnering with priority countries. Zambia hopes to achieve this goal by 2025 and interventions based on the identified hotspots should assist in this effort.

## Supporting information

**S1 STROBE Checklist.**
(DOCX)

**S1 Data.**
(CSV)

## Acknowledgments

We are grateful to the staff of the Ministry of Health, Zambia for allowing us to access data from their surveillance system database. Support rendered during data collection by colleagues at Centre for Infectious Diseases Research in Zambia -Enteric Diseases and Vaccine Research Unit.

## Author Contributions

**Conceptualization:** John Mwaba, Amanda K. Debes, Caroline Chisenga, Michelo Simuyandi, David Sack, Roma Chilengi, Mohammad Ali.

**Data curation:** John Mwaba, Amanda K. Debes, Patrick Shea, Orbrie Chewe, Mohammad Ali.

**Formal analysis:** John Mwaba, Amanda K. Debes, Patrick Shea, Mohammad Ali.

**Funding acquisition:** David Sack.

**Investigation:** John Mwaba, Amanda K. Debes, Patrick Shea, Mohammad Ali.

**Methodology:** John Mwaba, Amanda K. Debes, Patrick Shea, Mohammad Ali.

**Project administration:** Amanda K. Debes, David Sack, Roma Chilengi.

**Resources:** Amanda K. Debes, Victor Mukonka, David Sack, Roma Chilengi.

**Software:** Mohammad Ali.

**Supervision:** Amanda K. Debes, Roma Chilengi, Mohammad Ali.

**Validation:** John Mwaba, Amanda K. Debes, Geoffrey Kwenda, Roma Chilengi, Mohammad Ali.

**Visualization:** John Mwaba, Amanda K. Debes, Patrick Shea, Mohammad Ali.

**Writing – original draft:** John Mwaba, Amanda K. Debes, Patrick Shea, Mohammad Ali.

**Writing – review & editing:** John Mwaba, Amanda K. Debes, Patrick Shea, Victor Mukonka, Orbrie Chewe, Caroline Chisenga, Michelo Simuyandi, Geoffrey Kwenda, David Sack, Roma Chilengi, Mohammad Ali.

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
