## [Decision Letter · Decision Letter 0]

2 Dec 2019

Dear Dr. Ali:

Thank you very much for submitting your manuscript "Identification of cholera hotspots in Zambia: A spatiotemporal analysis of cholera data from 2008 to 2017" (#PNTD-D-19-00964) for review by PLOS Neglected Tropical Diseases. Your manuscript was fully evaluated at the editorial level and by independent peer reviewers. The reviewers appreciated the attention to an important problem, but raised some substantial concerns about the manuscript as it currently stands. These issues must be addressed before we would be willing to consider a revised version of your study. We cannot, of course, promise publication at that time.

We therefore ask you to modify the manuscript according to the review recommendations before we can consider your manuscript for acceptance. Your revisions should address the specific points made by each reviewer. 

When you are ready to resubmit, please be prepared to upload the following:

(1) A letter containing a detailed list of your responses to the review comments and a description of the changes you have made in the manuscript.

(2) Two versions of the manuscript: one with either highlights or tracked changes denoting where the text has been changed (uploaded as a "Revised Article with Changes Highlighted" file); the other a clean version (uploaded as the article file).

(3) If available, a striking still image (a new image if one is available or an existing one from within your manuscript). If your manuscript is accepted for publication, this image may be featured on our website. Images should ideally be high resolution, eye-catching, single panel images; where one is available, please use 'add file' at the time of resubmission and select 'striking image' as the file type. 

Please provide a short caption, including credits, uploaded as a separate "Other" file. If your image is from someone other than yourself, please ensure that the artist has read and agreed to the terms and conditions of the Creative Commons Attribution License at http://journals.plos.org/plosntds/s/content-license (NOTE: we cannot publish copyrighted images). 

(4) If applicable, we encourage you to add a list of accession numbers/ID numbers for genes and proteins mentioned in the text (these should be listed as a paragraph at the end of the manuscript). You can supply accession numbers for any database, so long as the database is publicly accessible and stable. Examples include LocusLink and SwissProt.

(5) To enhance the reproducibility of your results, we recommend that you deposit your laboratory protocols in protocols.io, where a protocol can be assigned its own identifier (DOI) such that it can be cited independently in the future. For instructions see http://journals.plos.org/plosntds/s/submission-guidelines#loc-methods

While revising your submission, please upload your figure files to the Preflight Analysis and Conversion Engine (PACE) digital diagnostic tool, https://pacev2.apexcovantage.com/ PACE helps ensure that figures meet PLOS requirements. To use PACE, you must first register as a user. Then, login and navigate to the UPLOAD tab, where you will find detailed instructions on how to use the tool. If you encounter any issues or have any questions when using PACE, please email us at figures@plos.org.

We hope to receive your revised manuscript by Jan 31 2020 11:59PM. If you anticipate any delay in its return, we ask that you let us know the expected resubmission date by replying to this email.

To submit a revision, go to https://www.editorialmanager.com/pntd/ and log in as an Author. You will see a menu item call Submission Needing Revision. You will find your submission record there. 

Sincerely,

Adam Akullian, Ph.D.

Associate Editor

Hélène Carabin

Deputy Editor

Please also address the following re: case definition on lines 105-109. The case definition is unclear. Please describe the inclusion / exclusion criteria for cases, including age-restrictions and whether cases needed to be culture confirmed or just "suspected." Also, it appears that patients under 5 were not included in the study. Please explain and consider adding this as a limitation. The word, "should" on line 108 suggests the case definition may have not been followed. Please comment.

Reviewer's Responses to Questions

**Key Review Criteria Required for Acceptance?**

**Methods**

-Are the objectives of the study clearly articulated with a clear testable hypothesis stated?

-Is the study design appropriate to address the stated objectives?

-Is the population clearly described and appropriate for the hypothesis being tested?

-Is the sample size sufficient to ensure adequate power to address the hypothesis being tested?

-Were correct statistical analysis used to support conclusions?

-Are there concerns about ethical or regulatory requirements being met?

Reviewer #1: This is a straightforward study with clearly defined aims and appropriate statistical methodologies employed. 

The cholera data section does not explicitly state the spatial resolution of the confirmed/suspected cases until the ethical section. Would it be possible to combine these sections or have the ethical section come immediately after? The cholera data feels lacking in detail about the spatial resolution of the data otherwise.

It would also be helpful if the section on population factors (lines 111-115) described the variables used in more detail rather than requiring the reader to wait for the table in the results section.

Reviewer #2: The objectives of the study was clearly stated, however the data sources and statistical analysis are not well described.

Are the cholera data used in this study monthly or yearly and at district level? Please explicitly state the level(s) in which the data was collected.

The authors provided a generic description of Zero Inflated Negtative Binomial model but did not explicitly relate the model to the current study. They mentioned that “y” is the independent variable. They should clearly state whether it is the counts of cholera cases in each district at a particular time (yearly?). In formulating the model, subscript should be used to allow the reader follow the methods easily. 

Line 137: Provide reference for SaTScan.

**Results**

-Does the analysis presented match the analysis plan?

-Are the results clearly and completely presented?

-Are the figures (Tables, Images) of sufficient quality for clarity?

Reviewer #1: The analysis of results is equally straightforward as the methods, although I have a few questions:

It is unclear what the numbers on the bars represent in Figure 1, as they do not seem to match the numbers mentioned in the preceding text.

The peak in 2009/2010 is striking. Do the authors have access to enough temporally refined data to study why this was the case, in terms of environmental/socio-economic factors? It would be helpful at least for the authors to discuss potential underlying reasons for this peak and then relative decline.

This leads me to also wonder if the hotspots found might differ if the authors split the data into, say, two groups - pre- and post-2010 and then re-ran their analysis on these two different time periods.

Figure 5 is of somewhat low resolution/quality.

Reviewer #2: The annotation in Figure 1 is not clear. What does the number on each bar represents? It does not correspond to the numbers on the y axis.

The maps in Figure 2 are too small. Change the format to 2 maps per row (5 by 2).

Table 2: Why is N=72? Since your study is spatiotemporal, you should have more data points than 72 districts times time. Were the 2008-2017 data sets aggregated for ZINB model?

On line 261, the authors said that “Distance from waterbodies” was the only significant variable. I will be cautious in reporting this as such, the confidence interval in Table 3 did not support this claim.

**Conclusions**

-Are the conclusions supported by the data presented?

-Are the limitations of analysis clearly described?

-Do the authors discuss how these data can be helpful to advance our understanding of the topic under study?

-Is public health relevance addressed?

Reviewer #1: Conclusions and limitations are clearly presented and supported by the study, so I have no major comments here.

Reviewer #2: The limitation should also include the use of a single (year) survey, ZDHS 2014 to extract the WASH data when the study period is 2008-2017.

**Editorial and Data Presentation Modifications?**

Reviewer #1: Some editorial modifications are needed throughout, where words are missing for example.

Reviewer #2: The manuscript "Identification of cholera hotspots in Zambia: A spatiotemporal analysis of cholera data from 2008 to 2017” will require full language edits. There are grammatical and orthographic errors here and there.

**Summary and General Comments**

Reviewer #1: As per my comments above, I find this to be a straightforward study which is of use for Zambian public health efforts to reduce the burden of cholera. My suggestions have been mainly to explore other ways of dividing the data to see if conclusions remain the same. It would be very interesting to see future work include data from the surrounding countries and look into human movement patterns across and within national boundaries to see how these are affecting the persistence of cholera in the region as a whole.

Reviewer #2: My major concern with this manuscript is the statistical methods used. Rather than using a range of regression models, a multilevel model (ZINB or ZIP with random effects) will suffice. The spatial heterogeneity of the cholera outbreaks will be captured by the random effects. Alternatively, a spatial ZINB or ZIP can be used with a spatially correlated errors. 

See for example, Loquiha O, Hens N, Chavane L, Temmerman M, Osman N, Faes C, Aerts M. Mapping maternal mortality rate via spatial zero-inflated models for count data: A case study of facility-based maternal deaths from Mozambique. PloS one. 2018 Nov 9;13(11):e0202186.

PLOS authors have the option to publish the peer review history of their article (what does this mean?). If published, this will include your full peer review and any attached files.

Reviewer #1: No

Reviewer #2: No

---

## [Decision Letter · Decision Letter 1]

17 Feb 2020

Dear Dr. Ali,

Thank you very much for submitting your manuscript "Identification of cholera hotspots in Zambia: A spatiotemporal analysis of cholera data from 2008 to 2017" for consideration at PLOS Neglected Tropical Diseases. As with all papers reviewed by the journal, your manuscript was reviewed by members of the editorial board and by several independent reviewers. The reviewers appreciated the attention to an important topic. Based on the reviews, we are likely to accept this manuscript for publication, providing that you modify the manuscript according to the review recommendations. 

Sincerely,

Adam Akullian, Ph.D.

Associate Editor

Hélène Carabin

Deputy Editor

Reviewer's Responses to Questions

**Key Review Criteria Required for Acceptance?**

**Methods**

-Are the objectives of the study clearly articulated with a clear testable hypothesis stated?

-Is the study design appropriate to address the stated objectives?

-Is the population clearly described and appropriate for the hypothesis being tested?

-Is the sample size sufficient to ensure adequate power to address the hypothesis being tested?

-Were correct statistical analysis used to support conclusions?

-Are there concerns about ethical or regulatory requirements being met?

Reviewer #1: The objectives of the study are clearly stated and straightforward, and the types statistical analyses are appropriate. However, I have a few questions and concerns about the data and study design:

Cholera data: What effect might the inclusion of suspected cases have on the analysis, particularly if the goal of the study is to identify vaccine targets rather than zones where sanitation improvements are needed? Are there not other water-borne pathogens in the country?

It also isn’t clear why cases were included for patients 5 years or older from areas where cholera was not known to have occurred. What is the justification behind this decision?

Population and socioeconomic data: In the introduction, the authors state that it is particularly peri-urban areas that are at highest risk for cholera transmission in Zambia. Would it be possible to further refine urban/rural classifications as such? It seems it would be important to identify and target not only those districts which are at risk, but specifically the peri-urban zones within these districts.

**Results**

-Does the analysis presented match the analysis plan?

-Are the results clearly and completely presented?

-Are the figures (Tables, Images) of sufficient quality for clarity?

Reviewer #1: Figure 1 – Does this figure represent confirmed cases alone, or both confirmed and suspected? It would be good to distinguish between the two in the figure. Furthermore, it seems as though cholera rates have declined overall since the peak in 2010. Can the authors comment on the reasons underlying this pattern? Given this pattern, it would be interesting to see analyses that focus on the data from recent years and compare it to the distribution of previous years. Have the patterns changed, or do they remain the same as for the peak years? If they have changed, what are the implications for vaccine targeting?

Figure 2 -There appears to be a colour in the map that is not represented in the legend (green). 

Results – paragraph beginning Line 263: It is interesting to see that some wards were not hotspots within districts, and is pertinent to my comment about identifying peri-urban areas above. Were these wards peri-urban?

**Conclusions**

-Are the conclusions supported by the data presented?

-Are the limitations of analysis clearly described?

-Do the authors discuss how these data can be helpful to advance our understanding of the topic under study?

-Is public health relevance addressed?

Reviewer #1: Conclusions are well supported by the data, and limitations are addressed albeit briefly. There is clear public health relevance of this study, but maps showing specific wards that should be targeted for vaccines would make this even clearer. Especially if the goal is specifically to focus limited vaccine resources.

**Editorial and Data Presentation Modifications?**

Reviewer #1: There are minor edits needed throughout for missing words and clarity.

**Summary and General Comments**

Reviewer #1: This paper is of public health interest for Zambia, and has implications for global cholera control as well. I believe it will be even stronger if analyses are split by time period (distinguishing recent years), and if greater attention is paid to ward-level outputs and peri-urban status.

PLOS authors have the option to publish the peer review history of their article (what does this mean?). If published, this will include your full peer review and any attached files.

Reviewer #1: No
---

## [Editor Report · Decision Letter 2]

17 Mar 2020

Dear Dr. Ali,

We are pleased to inform you that your manuscript 'Identification of cholera hotspots in Zambia: A spatiotemporal analysis of cholera data from 2008 to 2017' has been provisionally accepted for publication in PLOS Neglected Tropical Diseases.

Best regards,

Adam Akullian, Ph.D.

Associate Editor

Hélène Carabin

Deputy Editor

---

## [Editor Report · Acceptance letter]

27 Mar 2020

Dear Dr. Ali,

We are delighted to inform you that your manuscript, "Identification of cholera hotspots in Zambia: A spatiotemporal analysis of cholera data from 2008 to 2017," has been formally accepted for publication in PLOS Neglected Tropical Diseases.

Best regards,

Serap Aksoy

Editor-in-Chief

Shaden Kamhawi

Editor-in-Chief
